# STAND ON TWO SHOULDERS: DYNAMICALLY MERGING TOKENS FROM GENERAL AND MEDICAL EXPERTS

## ABSTRACT

In the realm of medical image analysis, the transferability of pre-trained Vision Transformers (ViTs) to specialized medical tasks remains a significant challenge. Previous approaches focus on adapting a single model, by introducing specialized learnable layers to the pre-trained model. However, a single model optimized for general tasks underperforms in domain-specific applications, while one medical models limited by their fundamental inferior capabilities, is not robust enough in real-world adaptation. To address this, we introduce the DynaMer Adapter, a novel architecture designed to enable Dynamically Merge tokens from general and medical pre-trained models, enhancing the adaptability of ViTs for medical imaging tasks. DynaMer incorporates a Gated Mixture-of-Expert (MoE) Adapter, ensuring that the model ingeniously prioritizes relevant features for specific medical tasks. Additionally, we incorporate a layer-wise skipping router within the architecture, designed to adjust the number of input tokens efficiently, thereby optimizing inference time without compromising on model accuracy. Extensive evaluations on the Medical Visual Task Adaptation Benchmark (Med-VTAB) demonstrate that DynaMer achieves state-of-the-art performance, particularly excelling in patient out-of-distribution settings and tasks with only few samples.

## 1 INTRODUCTION

The rapid advancement of deep learning in the field of medical image analysis has fostered some breakthroughs, yet the challenge of effectively transferring the knowledge from pre-trained models (He et al., 2021; Xie et al., 2021; Chen et al., 2021; Oquab et al., 2023) to specialized medical tasks persists. Vision Transformers (ViTs) (Dosovitskiy et al., 2021; Touvron et al., 2020; Liu et al., 2021) have shown exceptional performance in general image analysis tasks, and recently, there has been a lot of work exploring pre-training ViTs with medical images, thereby creating several models (Zhou et al., 2023; Huang et al., 2023; Xu et al., 2024). However, these models have not been widely adopted across different tasks as general domain pre-trained weights have. How to efficiently adapt pre-trained ViTs to medical downstream applications has not yet been widely explored.

Historically, adaptations of pre-trained ViTs to medical tasks (Jia et al., 2022; Yoo et al., 2023; Mo et al., 2024b) have involved the integration of specialized learnable layers or tokens. These modifications aim to tailor the model's focus towards features pertinent to medical images. However, this approach often struggles when directly applied some widely used weights (e.g., CLIP (Radford et al., 2021) or MAE (He et al., 2022)). The discrepancy arises from the fundamental differences in image characteristics and task requirements between general and medical imaging contexts. Another approach is to adopt ViTs pre-trained on medical images. This is also not ideal because the fundamental capabilities of medical pre-trained models are relatively inferior due to the limited data availability in the medical domain, making them not robust enough in real-world adaptation tasks. Moreover, these models are likely tailored to specific types of medical images, such as retinal images (Zhou et al., 2023) or pathology images (Xu et al., 2024). Identifying a pre-trained ViT that is suitable for downstream applications and demonstrates effective performance is a challenging task. This issue underscores a critical limitation: *a single model often fails to deliver optimal performance in specialized, domain-specific applications*.

To address these challenges, we introduce a novel architectural solution to effectively take advantage of pre-trained visual experts from both general and medical domains. We design an adapter to enable

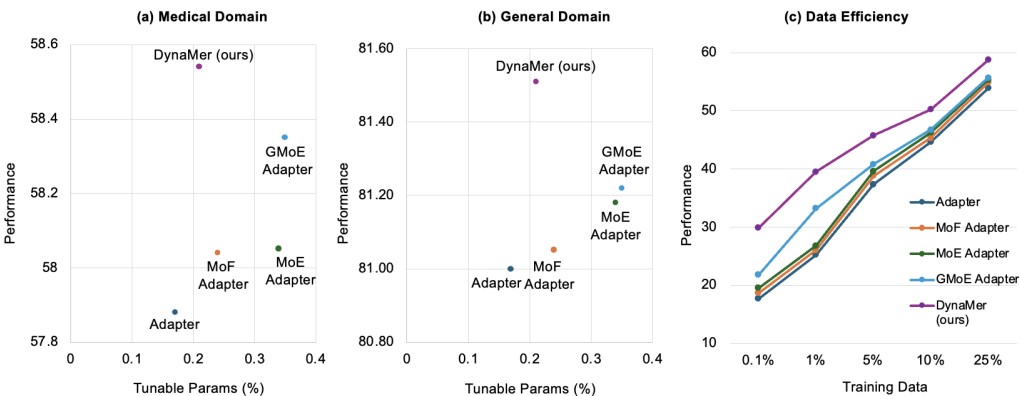

Figure 1: Illustration on the performance and computational efficiency of methods on (a) medical domain and (b) general domain. Our DynaMer achieves the best performance with only using few tunable parameters, demonstrating that DynaMer can effectively combine pre-trained models from general and medical domains while not causing high computational costs, which exactly meets the requirement in medical visual task adaptation. (c) Performance of various adapters with different amount of training data for adaptation, showing DynaMer has great data efficiency, which is critical for addressing the data scarcity issue in the medical domain.

Dynamically Merge tokens from general and medical ViTs (DynaMer Adapter), specifically for enhancing the adaptability of ViTs for a wide range of medical imaging tasks. DynaMer employs sophisticated layer-wise Mixture-of-Expert (MoE) adapters with gated mechanism that regulates the integration of tokens from general and medical domains, ensuring the model dynamically prioritizes the most relevant features for the task at hand.Beyond effectiveness in downstream tasks, we also notice that computational efficiency is critical for medical applications. On the one hand, since the gated MoE adapter in DynaMer learns the merging method according to information in each token, it is shared for both general and medical layers, thus only introducing few tunable parameters (see Figure 1), being efficient during training. On the other hand, we further introduce a layer-wise skipping router to strategically adjust the number of input tokens processed by the model. Together with the MoE mechanism, DynaMer largely reduce the inference time.

We evaluated our DynaMer Adapter through comprehensive testing on the Medical Visual Task Adaptation Benchmark (Med-VTAB) (Mo et al., 2024b), where it has demonstrated superior performance, setting new state-of-the-art results. Our evaluations specifically highlight the model's prowess in terms of both computation and data efficiency, illustrating its robustness and adaptability, as shown in Figure 1. Overall, our contributions can be summarized into four four folds:

- We introduce the DynaMer Adapter for medical visual task adaptation, a novel architecture that can Dynamically Merge tokens from general and medical pre-trained ViTs, to effectively take advantage of visual experts from both sides.

- DynaMer integrates layer-wise MoE adapters with gated mechanism. The sophisticated design enables deep merge of general and medical ViTs, making DynaMer outperforms traditional methods on a variety of medical imaging datasets, particularly in patient out-of-distribution scenarios and tasks with only few samples.

- DynaMer well emphasizes the critical need of reducing and managing computational costs in the medical domain. By incorporating the MoE mechanism and layer-wise skipping router, DynaMer achieves few costs in both training and inference time.

- Experimental results indicate that the principles underlying DynaMer, especially dynamically merging tokens from two pre-trained models in adaptation, could be extended beyond medical imaging to general domains requiring efficient and robust adaptation capabilities.

## 2  RELATED WORK

The development of the DynaMer Adapter is informed by several key areas of research, particularly in medical visual transfer learning, the use of adapters in medical contexts, and Mixture-of-Experts (MoE) models. This section outlines the seminal works and recent developments in these areas, highlighting both the motivation behind and the distinctions of our approach.

**Medical Visual Transfer Learning.** Transfer learning in medical imaging has seen significant interest (Rasmy et al., 2020; Wang et al., 2022; Xiao et al., 2023; Yang et al., 2023; Nguyen et al., 2023), particularly in adapting models trained on large, non-medical datasets to specific medical tasks. The utility of pre-trained Vision Transformers (ViTs) for such tasks has been explored extensively; however, these models often require careful tuning to overcome the domain shift between general and medical imaging datasets. Our work builds on this foundation by integrating a novel adaptation mechanism that leverages the strengths of Vision Transformers while addressing their limitations in domain-specific tasks. Furthermore, DynaMer is different from visual prompt tuning methods (Jia et al., 2022; Yoo et al., 2023; Mo et al., 2024b). DynaMer leverages a Gated Mixture-of-Experts (MoE) Adapter to dynamically integrate tokens from both general and medical pre-trained models, which allows the model to combine complementary knowledge from diverse domains. Unlike VPT, GaPT, and LSPT, which process all tokens through the transformer layers, DynaMer employs a router to skip less relevant tokens, reducing computational overhead while maintaining accuracy. DynaMer incorporates a gating network that intelligently balances contributions from the original and MoE-processed tokens, enhancing stability and task-specific adaptation.

**Medical Adapters.** Adapters (Pfeiffer et al., 2020) have become a popular method for tuning pre-trained models to new tasks without the need for extensive retraining. In the medical domain, adapters help mitigate the issues related to limited annotated medical data and significant domain-specific variations. Previous works have introduced adapters at different levels of neural architectures, focusing on efficiency and specificity. Our DynaMer extends this by incorporating a gating mechanism that dynamically manages the contributions of domain-specific adapters, enhancing both performance and adaptability. Compared to previous state-of-the-art methods, DynaMer introduces significant advancements in multiple dimensions. Unlike existing methods such as MoE (Shazeer et al., 2017) and GMoE (Mo et al., 2024a), which operate at the feature or layer level, DynaMer performs token-level integration. This enables finer granularity in combining features from general and medical pre-trained models, ensuring more effective task-specific adaptation. DynaMer introduces a dynamic gating network that balances contributions from original tokens and MoE-processed tokens. This mechanism improves stability during training and adapts to task-specific needs, especially in medical imaging, where feature priorities vary widely. While Cambrian-1 (Tong et al., 2024a) focuses on visual instruction tuning with LLMs, DynaMer targets medical image adaptation by combining two domain-specific models (general and medical) based on the layer-wise skipping router.

**Mixture of Experts Adapters.** The concept of Mixture-of-Experts (MoE) has been applied in various fields to improve model capacity and efficiency, primarily by routing different inputs to different 'expert' networks based on the input's characteristics. In the general domain, seminal works such as the exploration of feature mixtures and the recent study (Tong et al., 2024b) on the visual shortcomings of multimodal LLMs (Large Language Models) have highlighted the potential and challenges of MoE architectures. Our model adopts a similar motivational framework but diverges significantly in methodology by integrating a layer-wise, gated MoE structure that is specifically tailored for medical imaging tasks. This approach not only addresses the complexities inherent in medical image analysis but also contributes to the broader discourse on efficient and scalable model architectures. It should be emphasized that MoE experts are not trained and we are merging the advantage of pre-trained models from both general and medical domains. DynaMer is also different from Sparse-Gated MoE (Shazeer et al., 2017). Specifically, Sparse-Gated MoE operates at the layer or network level, activating a sparse subset of feed-forward networks (experts) per input. In contrast, DynaMer introduces token-level routing within each layer, enabling dynamic selection and processing of tokens based on their relevance to the task. While Sparse-Gated MoE focuses on improving the scalability and capacity of a single model, DynaMer is designed to fuse knowledge from two distinct pre-trained models by dynamically merging tokens from these two sources. Sparse-Gated MoE relies on a static gating network to determine which experts to activate. DynaMer, however, employs a dynamic gating mechanism that adjusts the balance between the original tokens and those processed by the MoE layer, ensuring task-specific stability and adaptability.

## 3 METHOD

In this section, we explore the methodology behind the DynaMer Adapter as shown in Figure 2, which is designed to enhance the adaptability and performance of pre-trained ViTs on specialized medical imaging tasks. We first provide preliminary concepts, followed by a detailed exposition of our novel adapter architecture.

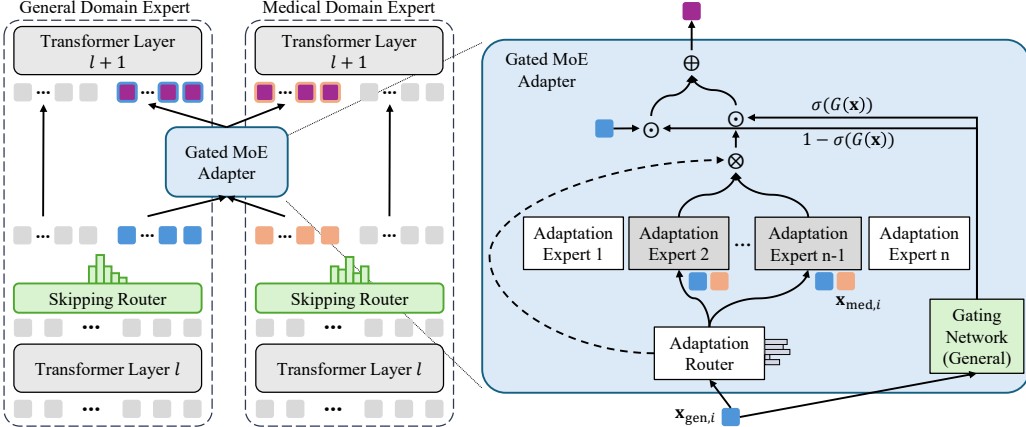

Figure 2: Illustration of the proposed DynaMer Adapter framework. Our DynaMer Adapter dynamically combines knowledge from both general and medical pre-trained models, where each layer utilizes a gated MoE adapter that decides the contribution of each domain-specific transformer block to the final task. The right figure shows how the gated MoE adapter processes a general token as an example. DynaMer also introduces a layer-wise skipping router that adjusts the number of input tokens from each layer to increase model efficiency. With these designs, DynaMer can dynamically allocate necessary capacities from diverse pre-trained experts for downstream medical visual tasks. Meanwhile, the computational efficiency, which is critical for the medical domain, is well addressed by DynaMer for both training and inference time.

## 3.1 PRELIMINARIES

Given a set of images, our target is to efficiently adapt pre-trained ViTs from medical and general domain to downstream medical visual tasks.

**Notations and Problem Setup.** Let $\mathbf{X} = [\mathbf{x}_1, \ldots, \mathbf{x}_N] \in \mathbb{R}^{N \times D}$ denote the input embedding tokens to a transformer, where $N$ is the number of tokens and $D$ is the embedding dimension. The transformer consists of $L$ layers, each comprising multi-headed attention (MSA) and a feed-forward network (FFN). Let $\mathbf{X}^{l-1}$ denote input tokens to the $l$-th transformer layer, so $\mathbf{X}^0 = \mathbf{X}$. Consider that there are two pre-trained ViTs, one from the general domain and another from the medical domain. Their weights are fixed during the adaptation stage. Tokens from the general model and the medical model are denoted by $\mathbf{X}_{\text{gen}}$ and $\mathbf{X}_{\text{med}}$, respectively.

**Revisit Adapter.** Adapters are small trainable modules inserted into pre-trained models, allowing for efficient fine-tuning on downstream tasks. Typically, these adapters consist of a bottleneck architecture, a down-projection followed by an up-projection, and are inserted in parallel with the FFN in each transformer layer. Most previous adapter-based methods are designed for a single model. Recently, Mo et al. (2024a) proposed a simple extension called MoE Adapter, making the adapter can be used for multiple pre-trained models. MoE Adapter combines tokens from two domains as:

$$\hat{\mathbf{x}}_i^l = \text{Adapter}^l([\mathbf{x}_{\text{gen},i}, \mathbf{x}_{\text{med},i}]). \tag{1}$$

Then, $\hat{\mathbf{X}}^l$ are sent to the next layer instead of original tokens. Here, Adapter are MLPs, and to produce different tokens for general and medical models, two separate MLPs are used as Adapter for the general and medical domains. Therefore, this method introduces relatively large number of tunalble parameters. Also, this method does not consider the specialization and dynamics for each token, leading to insufficient mining on designs of effectively combining two models.

## 3.2 GATED MIXTURE-OF-EXPERTS ADAPTER

The core innovation of DynaMer is the Gated Mixture-of-Experts (MoE) Adapter, which dynamically combines knowledge from both general and medical pre-trained models. The Gated MoE Adapter contains two designs: the MoE adapter and the gating mechanism. MoE adapter dynamically combines tokens from two pre-trained models according to the current token. This MoE adapter is shared for general and medical models, keeping the tunable parameter set small even two pre-trained models are involved during adaptation. Gating mechanism further balances the information from tokens processed by the adapter and the original tokens based on the information in the current

token. Each layer of the ViT incorporates a Gated MoE Adapter. For simplicity, we omit the layer superscript in this section.

**MoE Adapter.** Here, multiple adaptation expert networks $(\text{AdapE}_1, \text{AdapE}_2, \ldots, \text{AdapE}_n)$ are introduced, and an adaptation router network (AdapR) decided outputs of which experts should be used. Each adaptation expert can act as an adapter layer, taking two tokens from general and medical models, and output an integrated one. The adaptation router will take the current token as input and make decision according to it. It activates the top-$k$ expert networks with the largest scores. In order to sparsely activate different experts, the number of selected experts $k$ is fixed during training and much smaller than the total number of experts $n$. Taking a token from general domain as an example, the expert distribution of our MoE Adapter layers can be formulated as:

$$\text{AdapR}(\mathbf{x}_{\text{gen},i}) = \text{Softmax}(\text{KeepTopK}(\mathcal{R}_{\text{A}}(\mathbf{x}_{\text{gen},i}), k))$$

$$\tilde{\mathbf{x}}_{\text{gen},i} = \sum_{j=1}^{k} \text{AdapR}(\mathbf{x}_{\text{gen},i})_j \text{AdapE}_j(\mathbf{x}_{\text{gen},i}, \mathbf{x}_{\text{med},i}) \tag{2}$$

where $\text{AdapE}_j(\mathbf{x}_{\text{gen},i}, \mathbf{x}_{\text{med},i})$ denotes the output of the expert $\text{AdapE}_j$ when combining $i$-th tokens, and $\mathcal{R}_{\text{A}}(\cdot)$ is a learnable MLP within the router AdapR. KeepTopK is an operator to select the top $k$ ranked elements with the largest scores from output of $\mathcal{R}_{\text{A}}(\cdot)$, and only keep these values.

**Gating Mechanism.** After adding a randomly initialized adapter to each layer, we observe that the model turns to be unstable during training. This issue comes from the value distribution shift caused by inserting an random layer. Therefore, we further design a gating mechanism to dynamically balance tokens processed by the adapter and the original tokens, aiming at mitigating the above issue. Specifically, DynaMer introduce a learnable gating network $G$, which takes $\mathbf{x}$ as the input and outputs a gating vector. Then, the gating vector is used as weights to combine the original $\mathbf{x}$ and the processed $\tilde{\mathbf{x}}$. The process can be formulated as:

$$\hat{\mathbf{x}}_{\text{gen},i} = \sigma(G(\mathbf{x}_{\text{gen},i})) \cdot \tilde{\mathbf{x}}_{\text{gen},i} + (1 - \sigma(G(\mathbf{x}_{\text{gen},i}))) \cdot \mathbf{x}_{\text{gen},i} \tag{3}$$

where $\hat{\mathbf{x}}_{\text{gen},i}$ is the output of the adapter, $G$ is a trainable gating network, and $\sigma$ denotes the sigmoid function, ensuring that the gate outputs a value between 0 and 1. We use separate gating networks for general and medical model, while keeping them lightweight and only having one fully connected layer. The separate networks are used to capture different information propagation among layers in different pre-trained experts, and the lightweight design is for computational efficiency.

### 3.3 LAYER-WISE SKIPPING ROUTER

To further enhance model efficiency, especially for inference, we introduce a layer-wise routing mechanism that adjusts the number of input tokens which should be processed by the adapter based on the task's complexity and the specific medical imaging requirements. We omit the token source subscript in this section, because here, tokens are processed in the same way regardless of its source. The layer-wise skipping router and its usage can be formulated as:

$$\text{SkipR}(\mathbf{X}^l) = \text{TopKIndex}(\{\mathcal{R}_{\text{S}}(\mathbf{x}_i^l); i \in \{1, \cdots, N\}\}, k)$$

$$\hat{\mathbf{X}}^l = [\{\hat{\mathbf{x}}_i^l; i \in \text{SkipR}(\mathbf{X}^l)\}, \{\mathbf{x}_j^l; j \notin \text{SkipR}(\mathbf{X}^l)\}], \tag{4}$$

where $\mathcal{R}_{\text{S}}(\cdot)$ is a learnable MLP within the router SkipR. $\mathcal{R}_{\text{S}}(\cdot)$ takes every token in this layer as input and outputs a value indicating if this token needs to be sent to the adapter or not. TopKIndex is an operator to select the indexes of top $k$ ranked elements with the largest scores from the norm of feature outputs of the router $\mathcal{R}_{\text{S}}(\cdot)$ If the token $i$ is selected by the router from all tokens in this layer, this token will be sent to the gated MoE adapter, generating $\hat{\mathbf{x}}_i^l$. Otherwise, the token will skip the adapter. Processed tokens and skipped tokens are concatenated together, and sent to the next layer. Since general and medical pre-trained models may need different skipping mechanisms and this layer-wise router introduces few parameters, we use a separate layer-wise router network for two pre-trained experts.

At each layer, a router decides which tokens the gated MoE adapter should process, potentially reducing the number of tokens in deeper layers. This Layer-wise Skipping Router acts as a token-wise selection process to improve the computational efficiency for adaption tasks.This decision is based on

Table 1: Quantitative results of visual prompt tuning of DINO v2 pre-trained vision transformers on color images. Total Params denotes the total number of parameters for the backbone encoder ViT-B, prompt tokens or adapter parameters, and the task heads.

| Method | Total Params | HyperKvasir Polyp | MESAD Prostatectomy | AMLC Cell | APTOS Eye | ISIC Skin | Kvasir Polyp | LHNCBC Cell | MLLBone Cell | EyePACS Eye |
|---|---|---|---|---|---|---|---|---|---|---|
| Linear | 1.01X | 51.67 | 32.16 | 25.63 | 45.72 | 42.36 | 58.85 | 32.17 | 28.65 | 42.37 |
| VPT-Shallow (Jia et al., 2022) | 1.01X | 59.76 | 39.75 | 31.62 | 53.95 | 47.32 | 63.72 | 38.53 | 30.26 | 46.58 |
| VPT-Deep (Jia et al., 2022) | 1.04X | 62.89 | 43.78 | 35.75 | 57.52 | 50.89 | 66.53 | 42.87 | 35.37 | 48.75 |
| GaPT (Yoo et al., 2023) | 1.02X | 65.18 | 45.79 | 37.26 | 59.37 | 51.58 | 67.13 | 45.16 | 36.85 | 51.57 |
| LSPT (Mo et al., 2024b) | 1.05X | 67.23 | 47.53 | 38.72 | 61.25 | 53.62 | 69.79 | 47.51 | 38.92 | 52.86 |
| Adapter (Pfeiffer et al., 2020) (DINO v2) | 1.17X | 70.38 | 49.75 | 42.16 | 65.38 | 55.19 | 83.57 | 49.78 | 43.86 | 60.82 |
| Adapter (Pfeiffer et al., 2020) (Medical) | 1.17X | 70.29 | 49.72 | 42.37 | 65.23 | 55.02 | 83.35 | 50.26 | 44.25 | 60.78 |
| MoF-Adapter (Tong et al., 2024b) | 1.24X | 70.41 | 49.83 | 42.45 | 65.33 | 55.25 | 83.58 | 50.37 | 44.28 | 60.83 |
| MoE-Adapter (Mo et al., 2024a) | 1.34X | 70.42 | 49.78 | 42.47 | 65.39 | 55.21 | 83.58 | 50.35 | 44.32 | 60.89 |
| GMoE-Adapter (Mo et al., 2024a) | 1.35X | 70.75 | 50.26 | 42.83 | 65.51 | 55.37 | 83.79 | 50.86 | 44.75 | 61.02 |
| DynaMer Adapter (ours) | 1.21X | **70.82** | **50.53** | **43.08** | **65.73** | **55.53** | **83.92** | **51.07** | **45.03** | **61.15** |

the relevance of the information contained in each token, as assessed by the router, thus enabling the model to focus computational resources on the most informative parts of the input. This layer-wise adaptability not only speeds up the inference process but also reduces the computational load, making it feasible to deploy the model in real-time medical settings.

**Summary.** DynaMer's working process is summarized as follows. Once the sequence of tokens is produced by the previous transformer layer, the layer-wise skipping router looks into information in every token and then picks up the most relevant ones, sending them to the gated MoE adapter. One MoE adapter processes all tokens from both general and medical pre-trained models. This is achieved by the adaption router, which accordingly decides which adaptation experts should be activated based on every token. Particularly, although the $i$-th general token and medical token are processed together by the expert, they may activate different experts since their token information seen by the router is not the same. Moreover, a gating mechanism further offers balances between tokens processed by the MoE adapter and the original ones, by learning a gating network with the corresponding token as the input. Overall, DynaMer introduces these new learnable modules for adaptation: the shared MoE adapter containing a router and sparsely activated experts, the gating network, and the layer-wise skipping router. They are optimized end-to-end with the objective in adaptation tasks.

**Benefits.** The sophisticated and comprehensive designs in DynaMer offers several benefits for medical visual task adaptation. (1) All newly introduced modules by DynaMer are designed to dynamically process information according to the specific token. Such design fully considers the characteristics of the data and the pre-trained models during the adaptation phase, enabling different models to contribute the most relevant aspects to downstream tasks. DynaMer allows the general domain ViT to utilize its robust feature extraction capabilities, and medical domain ViT to leverage its specialized adaptability, boosting their ability to adapt to new, unseen medical data scenarios and culminating in a powerful tool for medical image analysis. (2) Since the MoE adapter already considers dynamics among data and models, we found that very lightweight adaptation experts can still achieve impressive results. Furthermore, as the MoE adapter is shared among general and medical models, DynaMer introduces much fewer new parameters compared to previous methods, greatly enhancing efficiency during training. (3) The layer-wise skipping router can dynamically determine which tokens can skip the adapter during inference, significantly reducing inference time.

## 4 EXPERIMENTS

### 4.1 EXPERIMENTAL SETUP

Our experiments are designed to rigorously evaluate the performance of DynaMer Adapter across a diverse set of medical imaging tasks. Below, we detail the datasets used, evaluation metrics, and implementation specifics.

**Datasets.** We utilize a comprehensive array of datasets in Med-VTAB (Mo et al., 2024a) to cover a broad spectrum of medical imaging challenges. These datasets are grouped into categories based on the image type: color medical images, X-ray images, and other modalities, including OCT, CT, and MRI. For color medical images, these nine datasets include images of polyps For X-ray images, these seven datasets address a variety of organs and conditions. For OCT, CT, and MRI modalities, these include seven datasets for the eye, chest, and brain. For the general domain, we use two widely used

Table 2: Quantitative results of visual prompt tuning of DINO v2 pre-trained vision transformers on X-ray images. Total Params denotes the total number of parameters for the backbone encoder ViT-B, prompt tokens or adapter parameters, and the task heads.

| Method | Total Params | Vindr Lung | CBIS Breast | COVIDx Lung | SYMH Shoulder | RSNA Bone Bone | CheXpert Chest | RSNA Lung |
|---|---|---|---|---|---|---|---|---|
| Linear | 1.01X | 62.81 | 71.32 | 72.56 | 72.81 | 46.73 | 67.26 | 65.38 |
| VPT-Shallow (Jia et al., 2022) | 1.01X | 63.56 | 72.23 | 73.83 | 74.35 | 50.21 | 69.73 | 67.69 |
| VPT-Deep (Jia et al., 2022) | 1.04X | 65.73 | 74.61 | 76.18 | 76.86 | 51.72 | 70.85 | 69.25 |
| GaPT (Yoo et al., 2023) | 1.02X | 66.92 | 75.15 | 77.25 | 77.25 | 52.83 | 71.37 | 70.29 |
| LSPT (Mo et al., 2024b) | 1.05X | 67.87 | 76.23 | 78.33 | 77.96 | 53.51 | 71.92 | 70.86 |
| Adapter (Pfeiffer et al., 2020) (DINO v2) | 1.17X | 70.35 | 81.26 | 80.72 | 79.52 | 55.35 | 73.61 | 72.93 |
| Adapter (Pfeiffer et al., 2020) (medical) | 1.17X | 70.25 | 81.32 | 80.76 | 79.46 | 55.29 | 73.58 | 72.91 |
| MoF-Adapter (Tong et al., 2024b) | 1.24X | 70.39 | 81.35 | 80.78 | 79.57 | 55.34 | 73.62 | 72.96 |
| MoE-Adapter (Mo et al., 2024a) | 1.34X | 70.37 | 81.35 | 80.82 | 79.56 | 55.36 | 73.63 | 72.95 |
| GMoE-Adapter (Mo et al., 2024a) | 1.35X | 70.62 | 81.67 | 81.15 | 79.78 | 55.47 | 73.68 | 73.05 |
| DynaMer Adapter (ours) | 1.21X | **70.86** | **82.15** | **81.87** | **80.56** | **55.93** | **74.52** | **73.86** |

Table 3: Quantitative results of visual prompt tuning of DINO v2 pre-trained vision transformers on OCT, CT, and MRI images. Total Params denotes the total number of parameters for the backbone encoder ViT-B, prompt tokens or adapter parameters, and the task heads.

| Method | Total Params | Heidelberg Eye | CC-CCII Chest | Mosmed Chest | COVID-C Chest | RICORD Chest | PPMI Brain | Brain-Tumor Brain |
|---|---|---|---|---|---|---|---|---|
| Linear | 1.01X | 63.25 | 60.87 | 62.87 | 60.93 | 58.35 | 55.27 | 62.35 |
| VPT-Shallow (Jia et al., 2022) | 1.02X | 64.15 | 60.75 | 63.21 | 61.05 | 59.07 | 56.35 | 62.75 |
| VPT-Deep (Jia et al., 2022) | 1.02X | 64.78 | 61.26 | 63.65 | 61.78 | 59.53 | 56.93 | 63.37 |
| GaPT (Yoo et al., 2023) | 1.02X | 65.06 | 61.37 | 63.69 | 61.95 | 59.71 | 56.97 | 63.52 |
| LSPT (Mo et al., 2024b) | 1.05X | 65.23 | 61.56 | 63.75 | 62.12 | 59.85 | 57.08 | 63.67 |
| Adapter (Pfeiffer et al., 2020) (DINO v2) | 1.17X | 67.58 | 66.23 | 65.52 | 66.37 | 64.21 | 61.35 | 67.62 |
| Adapter (Pfeiffer et al., 2020) (medical) | 1.17X | 67.53 | 66.25 | 65.58 | 66.39 | 64.22 | 61.36 | 67.68 |
| MoF-Adapter (Tong et al., 2024b) | 1.24X | 67.61 | 66.28 | 65.59 | 66.42 | 64.24 | 61.28 | 67.69 |
| MoE-Adapter (Mo et al., 2024a) | 1.34X | 67.65 | 66.26 | 65.56 | 66.38 | 64.25 | 61.39 | 67.70 |
| GMoE-Adapter (Mo et al., 2024a) | 1.35X | 67.76 | 66.43 | 65.68 | 66.51 | 64.42 | 61.46 | 67.73 |
| DynaMer Adapter (ours) | 1.21X | **68.23** | **66.89** | **66.21** | **66.97** | **64.82** | **61.86** | **68.15** |

classification datasets, FGVC and VTAB-1K. Following the prior work (Jia et al., 2022; Yoo et al., 2023; Mo et al., 2024b), we use the same split for training and validation.

**Evaluation Metrics.** To assess the effectiveness of our model, we employ a range of metrics that reflect both the accuracy and efficiency of medical image analysis. These metrics include, but are not limited to, classification accuracy, area under the ROC curve (AUC), and inference time.

**Implementation.** The DynaMer Adapter was implemented using PyTorch. Each expert within the MoE architecture was optimized individually before the gating mechanism was trained to dynamically combine their outputs. We fine-tuned the model on each dataset separately using Adam optimizer, with a learning rate of $1e-4$, and used the same pre-trained model parameters as previous work (Mo et al., 2024a). Specifically, we use DINO v2 (Oquab et al., 2023) general ViT-B/16 weights trained on 1.28 million general images and medical ViT-B/16 pre-trained weights (Nguyen et al., 2023) trained on 1.6 million cell images.

## 4.2 COMPARISON TO PRIOR WORK

To comprehensively assess the capabilities of our DynaMer Adapter, we performed extensive benchmarking against existing adaptation methods across various medical imaging modalities.

Table 1 shows our model outperforming traditional methods, particularly in complex cases like polyp detection and skin analysis. For X-ray images, as detailed in Table 2, our adapter provides significant improvements over existing methods, especially in distinguishing subtle features in chest and bone x-rays. In terms of OCT, CT, and MRI modalities, Table 3 highlights superior performance in modalities requiring high-detail orientation, such as brain tumor identification and chest CT analysis.

Our model demonstrated superior performance in adapting to diverse medical tasks, significantly outperforming baseline models across most metrics, particularly in challenging out-of-distribution scenarios. The results indicate that the dynamic and flexible nature of the proposed DynaMer Adapter provides a robust solution for medical visual task adaptation, addressing the limitations observed in previous models.

Table 4: Ablation results of Gated Mixture-of-Experts of general and medical pre-trained vision transformers on color images. Total Params denote the total number of parameters for the backbone encoder ViT-B, prompt tokens, and the task heads.

| General Gate | Medical Gate | Total Params | HyperKvasir Polyp | MESAD Prostatectomy | AMLC Cell | APTOS Eye | ISIC Skin | Kvasir Polyp | LHNCBC Cell | MLLBone Cell | EyePACS Eye |
|---|---|---|---|---|---|---|---|---|---|---|---|
| ✗ | ✗ | 1.19X | 70.38 | 49.82 | 42.56 | 65.32 | 55.28 | 83.65 | 50.52 | 44.36 | 60.86 |
| ✓ | ✗ | 1.20X | 70.55 | 50.23 | 42.68 | 65.41 | 55.36 | 83.72 | 50.78 | 44.53 | 60.93 |
| ✗ | ✓ | 1.20X | 70.67 | 50.36 | 42.85 | 65.56 | 55.42 | 83.81 | 50.82 | 44.62 | 60.98 |
| ✓ | ✓ | 1.21X | **70.82** | **50.53** | **43.08** | **65.73** | **55.53** | **83.92** | **51.07** | **45.03** | **61.15** |

Table 5: Ablation results of gated dimension of general and medical pre-trained vision transformers on color images. Total Params denote the total number of parameters for the backbone encoder ViT-B, adapter parameters, and the task heads.

| General Gate | Medical Gate | Total Params | HyperKvasir Polyp | MESAD Prostatectomy | AMLC Cell | APTOS Eye | ISIC Skin | Kvasir Polyp | LHNCBC Cell | MLLBone Cell | EyePACS Eye |
|---|---|---|---|---|---|---|---|---|---|---|---|
| 0 | 0 | 1.19X | 70.38 | 49.82 | 42.56 | 65.32 | 55.28 | 83.65 | 50.52 | 44.36 | 60.86 |
| 768 | 0 | 1.20X | 70.55 | 50.23 | 42.68 | 65.41 | 55.36 | 83.72 | 50.78 | 44.53 | 60.93 |
| 0 | 768 | 1.20X | 70.67 | 50.36 | 42.85 | 65.56 | 55.42 | 83.81 | 50.82 | 44.62 | 60.98 |
| 768 | 768 | 1.21X | **70.82** | **50.53** | **43.08** | **65.73** | **55.53** | **83.92** | **51.07** | **45.03** | **61.15** |
| 384 | 384 | 1.20X | 70.73 | 50.46 | 42.97 | 65.62 | 55.48 | 83.87 | 50.96 | 44.81 | 61.03 |
| 192 | 192 | 1.20X | 70.62 | 50.33 | 42.81 | 65.52 | 55.39 | 83.78 | 50.79 | 44.58 | 60.95 |
| 1 | 1 | 1.19X | 70.45 | 50.07 | 42.65 | 65.38 | 55.31 | 83.69 | 50.68 | 44.45 | 60.89 |

## 4.3 EXPERIMENTAL ANALYSIS

In this section, we delve deeper into the specific components and configurations of the DynaMer Adapter to understand their impact on performance. We present an ablation study on the gating mechanism, explore the effects of different gating dimensions and layers, and assess our model's performance in patient ID out-of-distribution scenarios and general domain adaptation.

**Ablation on Gated Mixture-of-Experts.** To evaluate the efficacy of the Gated Mixture-of-Experts mechanism, we conducted experiments where we systematically varied the number of experts and the complexity of the gating function. In Table 4, we compared these configurations against a baseline model without gating, measuring their impact on model accuracy and inference time across several medical imaging tasks. Our results indicate that the inclusion of the gating mechanism significantly improves the adaptability of the model to specialized tasks, confirming the hypothesis that dynamic feature routing enhances performance in domain-specific applications.

**Ablation on Gating Dimension.** We investigated the impact of different gating dimensions on the performance of the DynaMer Adapter, as shown in Table 5. By adjusting the dimensionality of the input to the gating network (dimensions tested: 768, 384, 192, 1), we assessed how this affects the model's ability to effectively combine the outputs of the experts. The experiments suggest an optimal range for the gating dimension that balances computational efficiency with task performance, providing insights into the model's sensitivity to this parameter.

**Ablation on Gating Layers.** In this study, we experimented with varying the number of layers equipped with the gating mechanism within the transformer architecture (layers tested: 12, 6, 3, 1). Our findings in Table 6 reveal that deeper integration of gating layers tends to yield better performance, particularly in complex imaging tasks, indicating that more extensive feature integration across layers enhances the model's effectiveness.

**Ablation on Layer-wise Skipping Router.** To further enhance model efficiency, especially for inference, we introduced a layer-wise routing mechanism that adjusts the number of input tokens based on the complexity of the task and the specific medical imaging requirements (ratios tested: 100%, 50%, 30%, 10%). Table 7 presents the effects of reducing the number of input tokens on computational efficiency and task performance. Our analysis demonstrates that strategic token reduction can significantly decrease inference time without substantially compromising performance, highlighting an effective trade-off between efficiency and accuracy.

**Patient ID Out-of-Distribution.** One of the critical evaluations of our model involved testing its performance on patient identification tasks where the test data distribution does not match the training data distribution, following the previous work (Mo et al., 2024a). This scenario tests the robustness

Table 6: Ablation results of gated layers of general and medical pre-trained vision transformers on color images. Total Params denote the total number of parameters for the backbone encoder ViT-B, adapter parameters, and the task heads.

| General Gate | Medical Gate | Total Params | HyperKvasir Polyp | MESAD Prostatectomy | AMLC Cell | APTOS Eye | ISIC Skin | Kvasir Polyp | LHNCBC Cell | MLLBone Cell | EyePACS Eye |
|---|---|---|---|---|---|---|---|---|---|---|---|
| 0 | 0 | 1.19X | 70.38 | 49.82 | 42.56 | 65.32 | 55.28 | 83.65 | 50.52 | 44.36 | 60.86 |
| 12 | 0 | 1.20X | 70.55 | 50.23 | 42.68 | 65.41 | 55.36 | 83.72 | 50.78 | 44.53 | 60.93 |
| 0 | 12 | 1.20X | 70.67 | 50.36 | 42.85 | 65.56 | 55.42 | 83.81 | 50.82 | 44.62 | 60.98 |
| 12 | 12 | 1.21X | **70.82** | **50.53** | **43.08** | **65.73** | **55.53** | **83.92** | **51.07** | **45.03** | **61.15** |
| 6 | 6 | 1.20X | 70.76 | 50.47 | 42.96 | 65.67 | 55.47 | 83.86 | 50.98 | 44.83 | 61.08 |
| 3 | 3 | 1.195X | 70.71 | 50.42 | 42.91 | 65.59 | 55.43 | 83.81 | 50.85 | 44.69 | 61.02 |
| 1 | 1 | 1.192X | 70.58 | 50.25 | 42.75 | 65.47 | 55.39 | 83.75 | 50.82 | 44.61 | 60.98 |

Table 7: Ablation results of Layer-wise Mixture-of-Experts tokens of general and medical pre-trained vision transformers on color images. Total Params denote the total number of parameters for the backbone encoder ViT-B, prompt tokens, and the task heads.

| # MoT tokens | Infer Time (s) per Batch | Total Params | HyperKvasir Polyp | MESAD Prostatectomy | AMLC Cell | APTOS Eye | ISIC Skin | Kvasir Polyp | LHNCBC Cell | MLLBone Cell | EyePACS Eye |
|---|---|---|---|---|---|---|---|---|---|---|---|---|
| 100% | 0.165 | 1.21X | 70.82 | 50.53 | 43.08 | 65.73 | 55.53 | 83.92 | 51.07 | 45.03 | 61.15 |
| 50% | 0.086 | 1.22X | **70.85** | **50.56** | **43.15** | **65.79** | **55.62** | **83.96** | **51.16** | **45.11** | **61.23** |
| 30% | 0.057 | 1.22X | 70.63 | 50.28 | 42.76 | 65.52 | 55.38 | 83.79 | 50.78 | 44.59 | 60.96 |
| 10% | 0.017 | 1.22X | 70.15 | 49.65 | 42.32 | 65.16 | 55.07 | 83.42 | 50.36 | 44.15 | 60.58 |

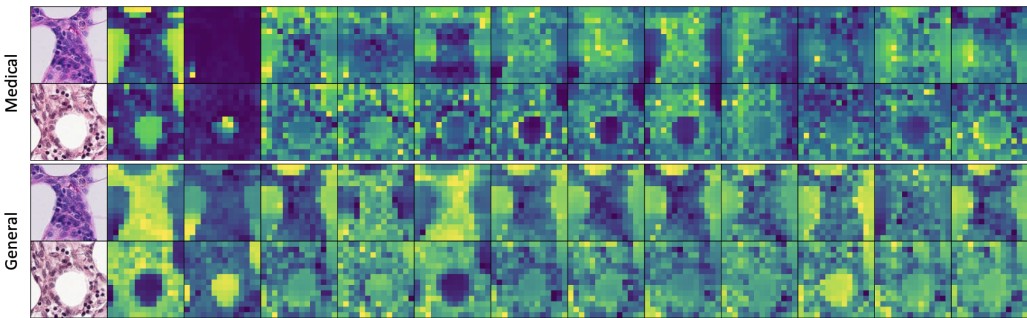

Figure 3: Qualitative visualization of attention maps learned by medical and general blocks in the proposed DynaMer Adapter.

of the model in real-world applications. Our DynaMer Adapter significantly outperformed traditional and other state-of-the-art methods, underscoring its robustness in handling out-of-distribution data, as detailed in Tables 8 and 9.

**General Domain Adaptation.** Furthermore, we assessed the capability of our model to adapt to general imaging tasks beyond the medical domain, employing the FGVC and VTAB-1K benchmarks, as shown in Table 10. This analysis helps us understand the versatility and broader applicability of the DynaMer Adapter. Despite its focus on medical imaging, preliminary results show promising adaptability, suggesting that the techniques developed could be extended to other domains of visual representation learning.

**Qualitative Visualization.** We also provide qualitative visualizations that illustrate how our model solves spatial and prompt forgetting problems typical of previous methods, as illustrated in Figure 3. For the task of using pathological slices to determine whether to transfer, the attention of previous methods in Figure 1 is sparse in the later layers, and the bright places may not correspond to cells. The attention of our method, however, can still accurately capture the position of cells, with bright spots that align well with cell locations, indicating active cell regions (empty signifies no cells; there is no attention). This contrast is significant as it highlights our DynaMer adapter's advanced capability of maintaining a focused and relevant feature representation across different layers of the transformer model. These visualizations not only demonstrate the efficacy of the DynaMer Adapter in maintaining focus on medically relevant features but also underscore its ability to enhance the interpretability of Vision Transformer models in medical applications.

Table 8: Patient ID Out-Of-Distribution results of our adapter vs. visual prompt tuning of non-medical pre-trained vision transformers on 160 patients. Total Params denote the total number of parameters for the ViT-B backbone, prompt tokens or adapter parameters, and the task heads.

| Method | Total Params | 160 Seen | 100 Seen | 60 Unseen | 80 Seen | 80 Unseen | 60 Seen | 100 Unseen |
|---|---|---|---|---|---|---|---|---|
| VPT-Shallow (Jia et al., 2022) | 1.01X | 38.53 | 38.42 | 38.35 | 38.37 | 38.29 | 38.25 | 38.13 |
| VPT-Deep (Jia et al., 2022) | 1.04X | 42.87 | 42.76 | 42.62 | 42.68 | 42.56 | 42.53 | 42.25 |
| GaPT (Yoo et al., 2023) | 1.02X | 45.16 | 45.06 | 44.92 | 44.95 | 44.82 | 44.76 | 44.32 |
| LSPT (Mo et al., 2024b) | 1.08X | 47.51 | 47.35 | 47.19 | 47.26 | 47.09 | 47.02 | 46.53 |
| Adapter (Pfeiffer et al., 2020) (DINO v2) | 1.17X | 49.78 | 49.56 | 49.63 | 49.47 | 49.68 | 49.38 | 49.57 |
| Adapter (Pfeiffer et al., 2020) (medical) | 1.17X | 50.26 | 50.16 | 50.21 | 50.08 | 50.23 | 49.95 | 50.16 |
| MoF-Adapter (Tong et al., 2024b) | 1.24X | 50.37 | 50.23 | 50.26 | 50.29 | 50.02 | 50.20 |
| MoE-Adapter (Mo et al., 2024a) | 1.34X | 50.35 | 50.21 | 50.23 | 50.12 | 50.26 | 50.01 | 50.17 |
| GMoE-Adapter (Mo et al., 2024a) | 1.35X | 50.86 | 50.53 | 50.58 | 50.36 | 50.62 | 50.21 | 50.42 |
| DynaMer Adapter (ours) | 1.21X | **51.07** | **50.89** | **50.93** | **50.78** | **50.97** | **50.72** | **50.95** |

Table 9: Patient ID Out-Of-Distribution results of our adapter vs. visual prompt tuning of non-medical pre-trained vision transformers on (a) 80 patients seen in the training set and (b) 20 patients unseen in the training set. Total Params denote the total number of parameters for the backbone encoder ViT-B, prompt tokens or adapter parameters, and the task heads.

| Method | Total Params | # Seen 80 | # Unseen 80 | # Unseen 60 | # Unseen 40 | # Unseen 20 |
|---|---|---|---|---|---|---|
| VPT-Shallow (Jia et al., 2022) | 1.01X | 38.37 | 38.29 | 38.25 | 38.21 | 38.27 |
| VPT-Deep (Jia et al., 2022) | 1.04X | 42.68 | 42.56 | 42.53 | 42.49 | 42.55 |
| GaPT (Yoo et al., 2023) | 1.02X | 44.95 | 44.82 | 44.78 | 44.73 | 44.79 |
| LSPT (Mo et al., 2024b) | 1.08X | 47.26 | 47.09 | 47.02 | 47.05 | 47.12 |
| Adapter (Pfeiffer et al., 2020) (DINO v2) | 1.17X | 49.47 | 49.68 | 49.72 | 49.69 | 49.75 |
| Adapter (Pfeiffer et al., 2020) (medical) | 1.17X | 50.08 | 50.23 | 50.28 | 50.23 | 50.32 |
| MoF-Adapter (Tong et al., 2024b) | 1.24X | 50.16 | 50.29 | 50.32 | 50.32 | 50.33 |
| MoE-Adapter (Mo et al., 2024a) | 1.34X | 50.12 | 50.26 | 50.23 | 50.25 | 50.28 |
| GMoE-Adapter (Mo et al., 2024a) | 1.35X | 50.36 | 50.62 | 50.56 | 50.59 | 50.63 |
| GL-MoE Adapter (ours) | 1.21X | **50.78** | **50.97** | **51.03** | **50.98** | **51.06** |

(a) 80 patients seen in the training set.

| Method | Total Params | 140 | 120 | # Seen 100 | 80 | 60 |
|---|---|---|---|---|---|---|
| VPT-Shallow (Jia et al., 2022) | 1.01X | 38.06 | 38.19 | 38.32 | 38.27 | 38.13 |
| VPT-Deep (Jia et al., 2022) | 1.04X | 42.15 | 42.47 | 42.63 | 42.55 | 42.49 |
| GaPT (Yoo et al., 2023) | 1.02X | 44.51 | 44.73 | 44.87 | 44.79 | 44.63 |
| LSPT (Mo et al., 2024b) | 1.08X | 46.73 | 46.95 | 47.19 | 47.12 | 47.01 |
| Adapter (Pfeiffer et al., 2020) (DINO v2) | 1.17X | 49.18 | 49.42 | 49.63 | 49.75 | 49.58 |
| Adapter (Pfeiffer et al., 2020) (medical) | 1.17X | 49.27 | 49.52 | 50.13 | 50.32 | 50.16 |
| MoF-Adapter (Tong et al., 2024b) | 1.24X | 49.75 | 49.96 | 50.12 | 50.33 | 50.21 |
| MoE-Adapter (Mo et al., 2024a) | 1.34X | 49.59 | 49.83 | 50.07 | 50.28 | 50.15 |
| GMoE-Adapter (Mo et al., 2024a) | 1.35X | 49.93 | 50.21 | 50.37 | 50.63 | 50.42 |
| GL-MoE Adapter (ours) | 1.21X | **50.52** | **50.65** | **50.91** | **51.06** | **50.97** |

(b) 20 patients unseen in the training set.

Table 10: Quantitative results of DINO v2 pre-trained vision transformers on FGVC and VTAB-1k datasets. Total Params denotes the total number of parameters for the backbone encoder ViT-B, prompt tokens or adapter parameters, and the task heads.

| Method | Total Params | CUB | Flowers | Cars | Dogs | NABirds | Nature | Specialized | Structured |
|---|---|---|---|---|---|---|---|---|---|
| VPT-Shallow (Jia et al., 2022) | 1.01X | 79.65 | 90.86 | 72.63 | 82.52 | 93.51 | 67.92 | 81.53 | 30.72 |
| VPT-Deep (Jia et al., 2022) | 1.04X | 83.02 | 94.85 | 79.56 | 83.71 | 76.35 | 70.64 | 83.26 | 42.65 |
| GaPT (Yoo et al., 2023) | 1.02X | 83.25 | 94.37 | 79.31 | 83.72 | 76.38 | 74.35 | 83.52 | 49.18 |
| LSPT (Mo et al., 2024b) | 1.08X | 84.37 | 95.23 | 80.28 | 84.37 | 77.28 | 77.32 | 85.82 | 52.93 |
| Adapter (Pfeiffer et al., 2020) (DINO v2) | 1.17X | 86.25 | 96.02 | 82.15 | 85.26 | 79.12 | 78.23 | 87.25 | 53.68 |
| Adapter (Pfeiffer et al., 2020) (CLIP) | 1.17X | 86.17 | 96.08 | 82.16 | 85.31 | 79.16 | 78.28 | 87.23 | 53.65 |
| MoF-Adapter (Tong et al., 2024b) | 1.24X | 86.29 | 96.12 | 82.21 | 85.32 | 79.19 | 78.33 | 87.26 | 53.69 |
| MoE-Adapter (Mo et al., 2024a) | 1.34X | 86.45 | 96.37 | 82.35 | 85.46 | 79.24 | 78.42 | 87.35 | 53.76 |
| GMoE-Adapter (Mo et al., 2024a) | 1.35X | 86.51 | 96.42 | 82.38 | 85.49 | 79.28 | 78.46 | 87.41 | 53.82 |
| DynaMer Adapter (ours) | 1.21X | **86.79** | **96.58** | **82.57** | **85.68** | **79.53** | **78.72** | **87.83** | **54.35** |

## 5 CONCLUSION

In this work, we present DynaMer Adapter, a novel Gated Layer-wise Mixture-of-Experts Adapter designed to enhance the adaptability and efficiency of pre-trained Vision Transformers (ViTs) for medical imaging tasks. Our approach addresses the significant challenge of transferring general visual learning to domain-specific tasks, particularly within the medical field where traditional transfer learning methods often fall short. The DynaMer Adapter integrates a sophisticated gated mechanism with a Mixture-of-Experts framework, allowing for dynamic adaptation based on the input data characteristics. This architecture not only tailors the processing pathways to specific tasks but also efficiently manages computational resources by adjusting the number of input tokens at each layer. Through extensive experimentation on a variety of medical datasets, our model demonstrated superior performance, especially in handling out-of-distribution data and patient identification tasks, setting new state-of-the-art benchmarks on the Medical Visual Task Adaptation Benchmark (Med-VTAB). Our work contributes to the ongoing discussions in the fields of medical visual transfer learning, adapter-based architectures, and Mixture-of-Experts models, highlighting the benefits and potential of our approach. Extensive empirical experiments and qualitative visualizations showcase the broader applicability of our methods to general domain adaptation, suggesting that the principles underlying the DynaMer Adapter could be extended beyond medical imaging.

## ETHICS STATEMENT

In accordance with the ICLR Code of Ethics, our research adheres strictly to ethical research standards. This study solely utilizes publicly available datasets within the medical imaging research community, ensuring that our work does not involve any private or personally identifiable information that could compromise individual privacy. While our DynaMer demonstrates significant potential for improving medical imaging analysis, we recognize the dual-use nature of AI technologies and the potential for misuse. We strongly advocate for the responsible application of our findings and encourage ongoing monitoring and regulation of AI applications in medical settings to prevent adverse outcomes. We are committed to engaging in discussions and receiving feedback to promote ethical usage and continuous improvement in AI-driven medical applications.

## REPRODUCIBILITY STATEMENT

We have detailed every aspect of our methodology to facilitate replication and verification by the broader research community. This includes an exhaustive description of experiments in Section **??** and comprehensive algorithmic details provided in Appendix B. For each experiment presented, we meticulously document the configurations, hyperparameters, and specific versions of the software used, which are detailed in Appendix C. Furthermore, to support the community in validating and building upon our work, we commit to making our codebase publicly available upon publication. This repository will include all necessary scripts, pre-trained models, and a step-by-step guide to re-running the experiments. By providing these resources, we aim to foster transparency and encourage future innovations inspired by our work.

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

APPENDIX

In this appendix, we provide the following material:

- addition implementation and datasets details in Section A,
- algorithm for our DynaMer Adapter in Section B,
- additional experimental analyses in Section C,
- additional qualitative visualization results in Section D,
- additional discussions on limitations and broader impact in Section E.

## A  IMPLEMENTATION & DATASET DETAILS

In this section, we provide detailed information on the implementation specifics and the diverse datasets incorporated into the Med-VTAB benchmark. These datasets cover a wide range of imaging modalities, including Color Images, X-ray, Optical Coherence Tomography (OCT), Computed Tomography (CT), and Magnetic Resonance Imaging (MRI). Each dataset is specifically chosen to reflect the diversity and complexity of medical visual tasks, facilitating robust and comprehensive model evaluation.

### A.1  COLOR IMAGES

Here, we detail datasets involving color images, each serving distinct medical applications, outlined with their respective number of images, unique characteristics, and medical relevance:

- **HyperKvasir** (Borgli et al., 2020): contains 110,079 images capturing polyps and other anatomical landmarks and pathological findings across 23 different classes.
- **MESAD Prostatectomy** (Bawa et al., 2021): comprises 29,454 images from prostatectomy procedures, with 21 different action classes for action classification in surgery.
- **AMLC** (Matek et al., 2019): includes 18,365 images of peripheral blood smears across 15 different morphological classes.
- **APTOS** (Aptos 2019 blindness detection): consists of 3,662 images rated for the severity of diabetic retinopathy on a scale from 0 to 4, across 5 classes.
- **ISIC** (Skin lesion images for melanoma classification): consists of 25,331 dermoscopic images for skin cancer classification across 9 diagnostic categories.
- **Kvasir** (Kvasirv2): contains 6,000 images for gastrointestinal cancer detection across 8 classes.
- **LHNCBC Malaria** (Lhncbc malaria): includes 27,560 images for malaria screening with 12 classes of cell annotations.
- **MLLBone** (Matek et al., 2021): consists of 171,374 images of blood cells across 21 different classes.
- **EyePACS** (Kaggle dr dataset (eyepacs)): contains 88,702 images for diabetic retinopathy classification into 5 severity levels.

### A.2  X-RAY

In this subsection, we outline X-ray datasets used in the benchmark, detailing the medical conditions addressed and the data volume:

- **Vindr** (Nguyen et al., 2022): contains 18,000 chest X-ray (CXR) scans identifying 14 critical findings.
- **CBIS-DDSM** (Lee et al., 2017): includes 10,239 mammographic images for breast cancer screening, categorized into normal, benign, and malignant findings.
- **COVIDx** (Wang et al., 2020): Comprises 194,922 images for COVID-19 detection, categorized into 4 classes.

- **SYMH** (Shoulder X-ray Classification): consists of 1,049 shoulder X-ray images across 4 categories.

- **RSNA Bone** (Halabi et al., 2019): contains 12,611 images for bone age assessment, spanning 228 age classes.

- **CheXpert** (Irvin et al., 2019): includes 224,316 chest radiographs for identifying conditions such as atelectasis, cardiomegaly, consolidation, edema, and pleural effusion.

- **RSNA** (Shih et al., 2019): consists of 29,684 chest radiographs, categorized into normal and pneumothorax positive.

## A.3   OCT, CT & MRI

We detail OCT, CT, and MRI datasets highlighting their specific applications and volume:

- **Heidelberg OCT** (Kermany et al., 2018): contains 84,495 OCT images across 4 categories related to eye diseases.

- **CC-CCII** (Zhang et al., 2020): includes 617,775 CT images focusing on COVID-19 pneumonia.

- **Mosmed** (Morozov et al., 2020): comprises 1,110 CT scans documenting COVID-19 pneumonia cases.

- **COVID-C** (Rahimzadeh et al., 2021): consists of 349 CT images for COVID-19 pneumonia detection.

- **RICORD** (Tsai et al., 2021): contains 120 CT images also focused on COVID-19 pneumonia.

- **PPMI** (Marek et al., 2011): includes 480 MRI scans related to Parkinson's disease.

- **Brain-Tumor** (Brain Tumor MRI Dataset): consists of 7,023 MRI images for brain tumor detection and segmentation.

For the general domain, we use two widely used classification datasets, FGVC and VTAB-1K. FGVC benchmark consists of 5 fine-grained classification tasks: CUB-200-2011 (Wah et al., 2011), Oxford Flowers (Nilsback & Zisserman, 2008), Stanford Cars (Gebru et al., 2017), Stanford Dogs (Khosla et al., 2011), and NABirds (Van Horn et al., 2015). Following the prior work (Jia et al., 2022; Yoo et al., 2023; Mo et al., 2024b), we use the same split for training and validation. VTAB-1K (Zhai et al., 2019) dataset includes 19 diverse visual classification tasks and three groups: Natural images obtained from standard cameras, Specialized images captured using specific equipment, and Structured images for object counting. Each task contains 1000 training samples, and we use the same split in (Jia et al., 2022; Yoo et al., 2023; Mo et al., 2024b) to run the final training and evaluation.

## B   ALGORITHM FOR DYNAMER ADAPTER

In this part, we outline the algorithm of the DynaMer Adapter, detailing the gating mechanism and how the layer-wise routing and mixture-of-experts are implemented to adapt to the input features dynamically. The DynaMer Adapter integrates a gating mechanism and multiple expert networks within the architecture of a Vision Transformer (ViT) to dynamically adapt to specific medical imaging tasks. Below, we describe the step-by-step operation of the adapter within a ViT layer.

Algorithm 1 integrates the DynaMer Adapter into each transformer layer. The adapter employs two expert networks, one tailored for general visual tasks ($E_g$) and the other for medical-specific tasks ($E_m$), each trained with their respective domain-specific pre-trained weights ($W_g$ and $W_m$). The gating network (GateNetwork) dynamically computes gating values for each token based on its embedded representation, controlling the contribution of each expert's output to the final layer output. The gating mechanism ensures that the model dynamically prioritizes relevant features for the task at hand, enhancing both specificity and adaptability.

---

**Algorithm 1** Algorithm for DynaMer Adapter

---

1: **Input:** Input token embeddings $X \in \mathbb{R}^{N \times D}$, where $N$ is the number of tokens and $D$ is the embedding dimension.
2: **Output:** Adapted output tokens $Y \in \mathbb{R}^{N \times D}$
3: **Initialization:**
4: Load pre-trained general domain weights $W_g$
5: Load pre-trained medical domain weights $W_m$
6: **for** each layer $l$ in ViT **do**
7:     Compute initial forward pass:
8:     $X' \leftarrow \mathrm{MSA}(X) + X$                       ▷ Multi-headed Self-Attention (MSA)
9:     $X'' \leftarrow \mathrm{FFN}(X') + X'$                   ▷ Feed-Forward Network (FFN)
10:    Initialize router $\mathcal{R}$ and gating network $G$
11:    Initialize experts $\mathcal{E}_1, \mathcal{E}_2, ..., \mathcal{E}_n$
12:    **for** each token $x_i$ in $X''$ **do**
13:       $E_g[i] \leftarrow \mathrm{ExpertGeneral}(x_i, W_g)$
14:       $E_m[i] \leftarrow \mathrm{ExpertMedical}(x_i, W_m)$
15:       Compute gating values:
16:       $g_i \leftarrow \sigma(\mathrm{G}(x_i))$                      ▷ Sigmoid function $\sigma$ for gating
17:       Combine expert outputs:
18:       $y_i \leftarrow g_i \cdot E_g[i] + (1 - g_i) \cdot E_m[i]$
19:    Apply layer-wise routing to adjust input tokens:
20:    $X \leftarrow \mathrm{LayerwiseRouter}(Y, \mathcal{R}, m)$       ▷ Select top $m$ tokens for next layer
21:    $Y \leftarrow \mathrm{Concatenate}(y_1, y_2, \ldots, y_N)$
22: **return** $Y$

---

## C   Additional Experimental Analyses

In this section, we provide further experimental analyses on patient ID out-of-distribution (OOD). One of the most rigorous tests for any model developed for medical applications is its performance on OOD data, particularly in scenarios where patient identification accuracy is crucial. This test is essential for assessing the robustness of the model under conditions that diverge from those seen during training. Our experiments on OOD performance were structured to evaluate how well the model could identify patient data it had either seen but under different conditions or had never seen before. We utilized a split of 160 patients, some of whom were part of the training dataset and others completely unseen during the training phase. This setup was designed to closely mimic real-world situations in which a model must generalize well beyond its training examples. As reported in Tables 8 and 9, the DynaMer Adapter outperforms both traditional methods and other state-of-the-art adapters in scenarios involving both seen and unseen patients. Specifically, our adapter demonstrates a notable increase in identification accuracy across all splits compared to baseline models and even other advanced adapters.

For the setting with 160 seen patients in Table 8, the DynaMer Adapter achieved an accuracy of 51.07%, which is significantly higher than the LSPT model (Mo et al., 2024b) and even outperforms other advanced adapters like GMoE-Adapter (Mo et al., 2024a). Furthermore, when it comes to more challenging settings with fewer seen patients (100 and 60 unseen), our model consistently showed less drop in performance, indicating robust feature extraction and generalization capabilities. Regarding 80 seen patients in Table 9, our model maintains high accuracy (50.78%) when 40 patients are unseen, which is an improvement over methods like VPT-Shallow and GaPT, illustrating the efficacy of our gating mechanism in handling OOD data. Even in the most challenging scenario where only 20 patients are seen, the DynaMer Adapter manages to outperform all other methods with accuracies above 50%, demonstrating the model's ability to leverage both general and domain-specific knowledge effectively. These results suggest that the gating mechanism within the DynaMer Adapter plays a crucial role in dynamically adjusting the contributions of the general and medical expert networks based on the input data. This dynamic adjustment is critical in OOD scenarios, as it allows the model to handle better the variability and unpredictability associated with unseen patient data.

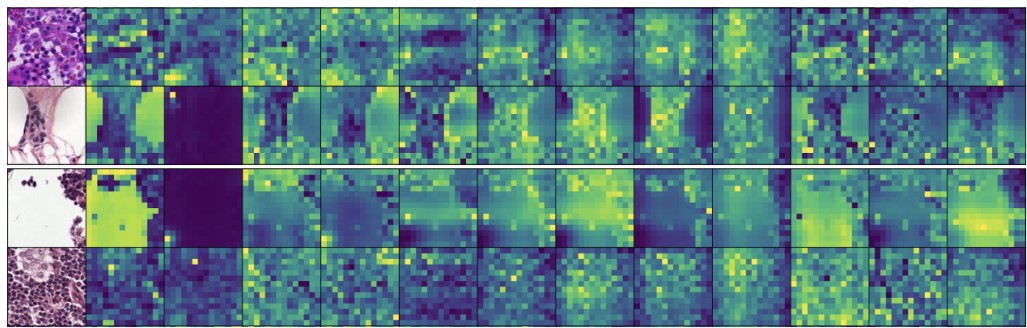

Figure 4: Qualitative visualization of attention maps learned by medical blocks in the proposed DynaMer Adapter.

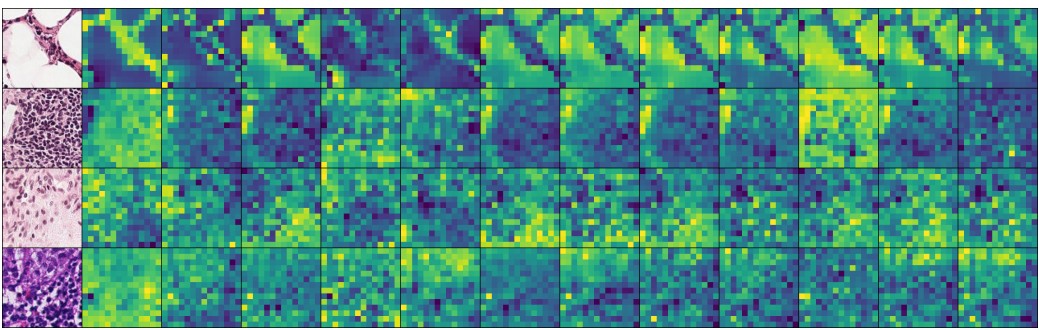

Figure 5: Qualitative visualization of attention maps learned by general blocks in the proposed DynaMer Adapter.

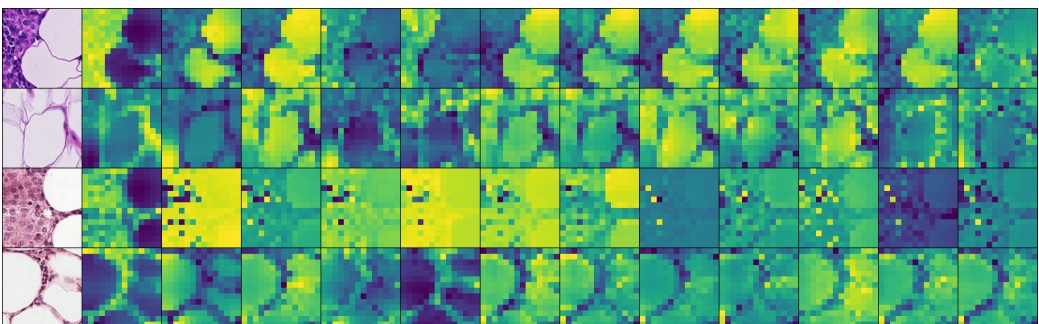

Figure 6: Qualitative visualization of attention maps learned by general blocks in the proposed DynaMer Adapter.

The superior performance of the DynaMer Adapter in OOD patient identification tasks underscores its potential for real-world medical applications. By effectively managing discrepancies between training and test distributions, our model ensures reliable performance, making it a valuable tool for scenarios where robustness to OOD data is paramount. This adaptability is particularly crucial in medical settings, where encountering unseen variations is common. To further enhance the OOD robustness, future work could explore more sophisticated routing mechanisms or deeper integration of domain-specific knowledge, potentially through semi-supervised learning techniques or unsupervised domain adaptation strategies to better capture and generalize across diverse patient data.

## D  ADDITIONAL QUALITATIVE VISUALIZATIONS

In this section, we include more qualitative visualizations to demonstrate the effectiveness of the DynaMer Adapter in handling complex visual tasks in medical imaging. These visualizations in

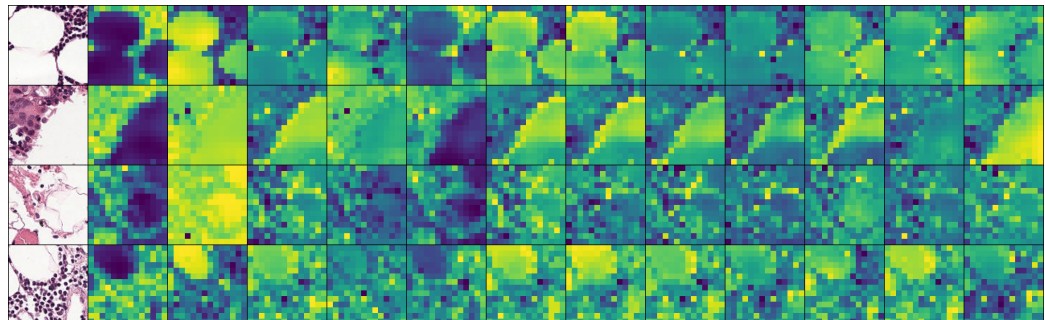

Figure 7: Qualitative visualization of attention maps learned by general blocks in the proposed DynaMer Adapter.

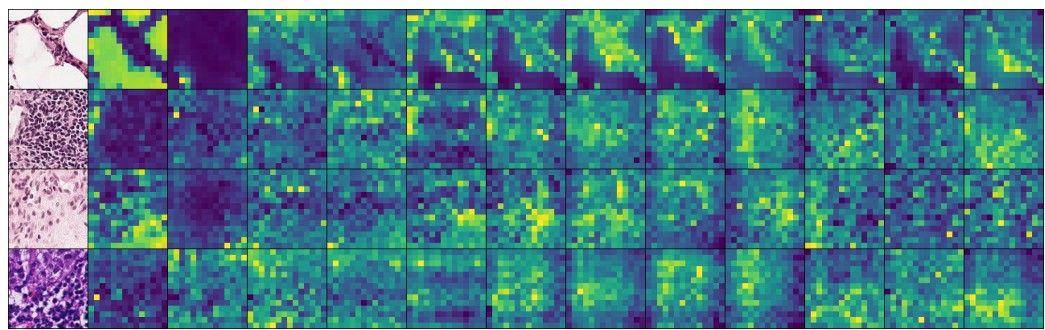

Figure 8: Qualitative visualization of attention maps learned by medical blocks in the proposed DynaMer Adapter.

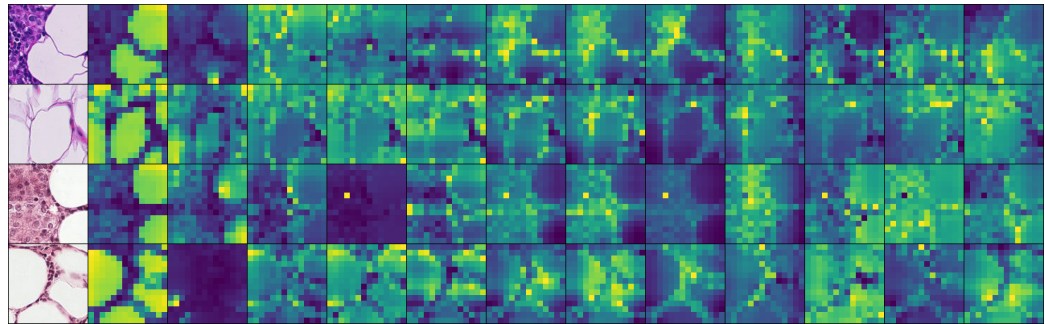

Figure 9: Qualitative visualization of attention maps learned by medical blocks in the proposed DynaMer Adapter.

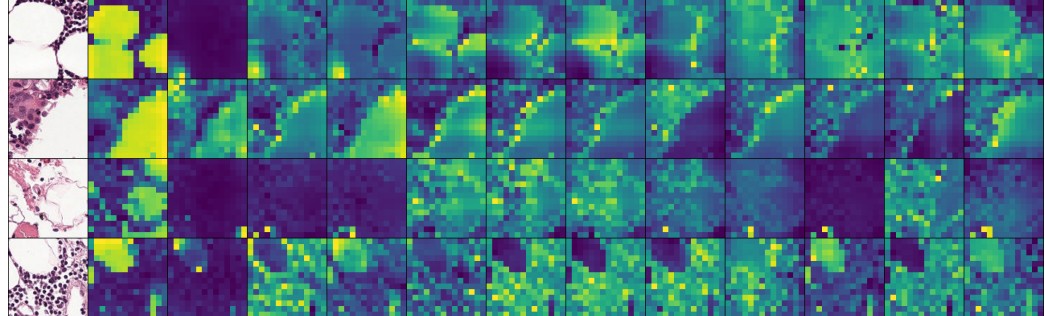

Figure 10: Qualitative visualization of attention maps learned by medical blocks in the proposed DynaMer Adapter.

Figure 4 illustrate how the model maintains focus on relevant features, addressing challenges such as spatial and prompt forgetting.

In addition to providing a robust solution to the problem of attention drift, where traditional methods lose focus on the relevant features in deeper layers or more complex scenarios, our visualizations show that the DynaMer Adapter effectively manages this issue. This is achieved by dynamically adjusting the influence of general and medical expert networks through our innovative gating mechanism. The adapter ensures that the model's attention mechanism remains relevant to the medical task at hand, irrespective of the inherent complexities or the variability of the medical images.

These visualizations in Figures 5, 6, and 7 alongside Figures 8, 9, and 10 demonstrate the distinctive characteristics of the attention maps in the general and medical blocks, validating the motivation and potential of our DynaMer Adapter. Particularly, the attention in the middle layers (layers 5-8) reveals differences in the properties of the attention maps between the two types of blocks. In the general block, the attention often focuses on the edges of tissue, possibly reflecting interactions with shapes and edges. Conversely, in the medical block, the attention frequently centers on cellular regions, especially areas with dense nuclei, likely due to the medical block's focus on tissue cell characteristics. This suggests that the dynamic adjustment and integration through the gating mechanism might be a plausible reason for the effectiveness of the DynaMer Adapter in handling complex visual tasks in medical imaging.

## E   MORE DISCUSSIONS

### E.1   LIMITATIONS

While the DynaMer Adapter showcases innovative advancements in adapting pre-trained Vision Transformers (ViTs) for specialized medical imaging tasks, it is important to acknowledge several inherent limitations:

- **Scalability Challenges:** While the DynaMer Adapter performs well on structured benchmarks, its scalability to extremely varied medical conditions without considerable customization remains untested. The computational demands may also escalate with the increase in the number of experts, potentially limiting its applicability in resource-constrained settings.

- **Generalization across Diverse Medical Tasks:** Although the adapter is designed to be adaptable, its performance may still depend on the similarity between the training scenarios and the target tasks. Variations in medical imaging data, such as differences in imaging techniques or pathology characteristics, could affect the model's ability to generalize effectively across tasks not seen during training.

- **Dependency on High-Quality Annotations:** The performance of the DynaMer Adapter is contingent on the availability of high-quality, annotated datasets. In medical imaging, where annotations require expert medical knowledge, the scarcity of detailed annotations can limit the training effectiveness and accuracy of the model.

### E.2   BROADER IMPACT

The development of the DynaMer Adapter has implications that extend beyond the field of medical imaging, influencing both societal norms and technological advancements:

- **Enhancement in Healthcare Quality:** By improving diagnostic accuracy and efficiency, the DynaMer Adapter has the potential to enhance patient care quality significantly. Faster and more accurate diagnostics can lead to better patient outcomes, particularly in conditions where early detection is crucial.

- **Economic Impact:** More efficient diagnostics could reduce the cost burden on healthcare systems by decreasing the need for repeat tests and speeding up the diagnosis process. However, the high costs associated with developing and implementing such advanced AI systems could also widen the gap in medical services between high and low-resource settings.

- **Ethical and Privacy Concerns:** The integration of AI in medical diagnostics raises substantial ethical and privacy concerns, especially regarding data handling, patient consent, and the potential biases in AI models. Ensuring that these technologies are developed and implemented responsibly is crucial to maintaining public trust.

- **Potential for Broader Applications:** The underlying principles of the DynaMer Adapter, including dynamic adaptation and efficient computational resource management, are applicable in other domains that deal with large-scale data and require robust, adaptable solutions. This includes areas like climate modeling, autonomous driving, and personalized education, where similar challenges in handling diverse, high-dimensional data are present.

Addressing both the limitations and recognizing the broader impacts are essential for guiding the future development and deployment of the DynaMer Adapter, ensuring it brings benefits while mitigating potential risks.

