# OpenReview forum: "Stand on Two Shoulders: Dynamically Merging Tokens from General and Medical Experts"
_ICLR.cc/2025/Conference — Submitted to ICLR 2025_

### Official Review · Reviewer_7oDt · 2024-10-27

**Soundness:** 3
**Presentation:** 3
**Contribution:** 2
**Rating:** 3
**Confidence:** 5

**Summary:**

The paper introduces the DynaMer Adapter, a novel architecture that enhances Vision Transformers' adaptability for medical imaging tasks by merging tokens from general and medical pre-trained models.

**Strengths:**

It features a Gated Mixture-of-Experts Adapter for prioritizing task-relevant features and a layer-wise skipping router for optimizing inference time. The DynaMer Adapter achieves state-of-the-art performance on the Med-VTAB benchmark, particularly in out-of-distribution patient settings and with limited samples. The paper demonstrates the potential for broader applicability of DynaMer's principles beyond medical imaging.

**Weaknesses:**

1. While the paper introduces the DynaMer Adapter by leveraging the concept of the Mixture-of-Experts (MoE) at both the feature and token levels, it's crucial to articulate the specific innovations beyond the existing MoE framework. The paper would benefit from a more detailed discussion on how the DynaMer Adapter's approach differs from current state-of-the-art methods, including references to related work that showcases the incremental advancement. Regarding the Layer-wise Skipping Router, clarifying its mechanism as a token-wise selection process could enhance understanding and emphasize its role in improving computational efficiency.

2. The paper's experimental section would be significantly strengthened by including comparisons that demonstrate the value of fusing general and medical pre-trained models over a task-specific, medically trained model. It's essential to show that the combined model not only adapts well but also surpasses the performance of a model trained solely on medical data. This could be achieved by designing experiments that benchmark the DynaMer Adapter against a medical model trained on the same tasks, highlighting the benefits of incorporating general domain knowledge.

**Questions:**

Computational Cost Analysis， flops GMAc

---

> ### Comment · Reviewer_7oDt · 2024-11-30
>
> Thank you for the detailed response!
>
> I still feel my first concern is not fully addressed, partially because the authors presented an alternative experiment than what I suggested.
>
> Overall, I am willing to raise the soundness score for the additional results but maintain my overall assessment.

---

### Official Review · Reviewer_2gYq · 2024-11-03

**Soundness:** 4
**Presentation:** 3
**Contribution:** 3
**Rating:** 5
**Confidence:** 4

**Summary:**

A single model optimized for general tasks often falls short in domain-specific applications. This paper presents the DynaMer Adapter, an architecture designed to dynamically merge tokens from both general and medical pre-trained models, thereby enhancing performance in downstream medical imaging tasks. It features a Gated Mixture-of-Experts (MoE) Adapter, which intelligently prioritizes relevant features for specific medical applications. Additionally, the authors introduce a layer-wise skipping router within the architecture. Evaluation results on several benchmarks indicate that DynaMer achieves outstanding performance, particularly in patient out-of-distribution scenarios and tasks with limited sample sizes.

**Strengths:**

The originality of the work is commendable. The authors propose a new solution to an existing topic. However, the limitations of prior work are not clearly presented, which the authors could further enhance.

**Weaknesses:**

1. The novelty of the proposed method is unclear.

1.1 The distinctions between this approach and existing methods such as MOE, MOF, GMOE, and Adapter need to be better articulated.
Additionally, some relevant works have not been discussed.
Regarding Cambrian-1: A Fully Open, Vision-Centric Exploration of Multimodal LLMs (https://arxiv.org/abs/2406.16860):

1.2 The proposed method appears to be similar to concepts presented in the paper
A Large-Scale Medical Visual Task Adaptation Benchmark, 2024. https://arxiv.org/abs/2404.12876
Both utilize gated MOE; what are the specific differences?

2. Furthermore, the performance gains of the proposed method are limited.

2.1 The improvements compared to existing approaches such as MOE, MOF, GMOE, and Adapter are minimal. As shown in Figure 1, the proposed method only achieves about a 0.5 improvement over MOF. How can it be claimed as effective in this field? The authors are encouraged to clarify the significance of the performance gains in relation to existing methods.

2.2 The effectiveness of the layer-wise skipping routers is difficult to verify in this paper. How can the authors demonstrate the effectiveness of this approach?

3. The proposed method is quite close to the following work; however, the author has not addressed the differences.

Outrageously Large Neural Networks: The Sparsely-Gated Mixture-of-Experts Layer, https://openreview.net/pdf?id=B1ckMDqlg, ICLR 2017.

**Questions:**

I am open to increasing my scores if the authors can address my comments above.

---

### Official Review · Reviewer_uQAj · 2024-11-04

**Soundness:** 3
**Presentation:** 3
**Contribution:** 2
**Rating:** 6
**Confidence:** 4

**Summary:**

The paper proposed a new mixture of expert mechanisms to combine pre-trained general- and medical-ViT models. The MoE algorithm includes key steps: (a) incorporating Gated Mixture-of-Expert to combine original tokens and tokens after MoE layers; (b) using a Skipping Router to select top-k relevant tokens for MoE components; (c) adapting MoE at each ViT layer as adaptor method.

Authors conduct a wide range of experiments on general and medical downstream tasks with fine-tuning. The paper shows improvement results on several datasets and outperforms several adaptor and MoE-based approaches.

**Strengths:**

Reviewers see the following strengths:

(a) Authors applied **layer-wise** MoE adaptor to merge features from general and medical ViT models, which is different from prior work based on block features of ViT.

(b) To further reduce computational costs, they proposed a *skipping layer* to select the top relevant tokens used for MoE while the remaining ones fed into the next layers. Furthermore, the idea of using the *gating network* to combine original tokens and output after MoE to make the model stable learning is also interesting.

(c) The experiments are diverse, covering several datasets with detailed ablation studies to support the proposed components in the paper (Gated Mixture-of-Experts, Gating Dimension, Layer-wise Skipping, etc.)

**Weaknesses:**

While the method is interesting and novel, the Reviewer is concerned about the significant improvements of experiments. For e.g.,
In Tables 1, 2, and 3, **DynaMer Adaptor** outperforms other MoE baselines with a *slight margin* (ranging from 0.5% to 1%) while the total parameter is higher than two others, e.g., Adapter with 1.17X.

In another task with the out-of-domain prediction (Table 9-b), the tasks usually indicate a large gap between baselines; *DynaMer Adaptor* only surpasses other MoE approaches with a similar margin as fine-tuning cases. Therefore, it seems to reviewers that most MoE baselines have similar performance, resulting in *DynaMer Adaptor*'s contributions not being really clear.

Reviewers would suggest authors conduct studies in more challenging settings, for e.g., zero-shot or few-shot with linear probing, to highlight the benefits of DynaMer Adaptor. Given these, the Reviewer would revise the rating.

**Questions:**

There are some points unclear to Reviewers:

(i) In equation (4), Which exact outputs does the TopKIndex take from $R_S(.)$ to choose a token for MoE? Is it based on the norm of feature outputs or some activation functions?

(ii) Intuitionly, designing a Skipping Router (SR) is not optimal yet. For e.g., there is no conditional information for *SR* to guide the model correctly on which tokens should be used in MoE and which ones should be used for the next layer. The information to update the *SR* course can be derived from gradient returns from the loss function, but the order of tokens used by *SR* has not yet been considered. So, do authors think integrating the **differentiable TopKIndex** will help improve accuracy?

---

### Official Review · Reviewer_R5Zb · 2024-11-04

**Soundness:** 3
**Presentation:** 2
**Contribution:** 2
**Rating:** 5
**Confidence:** 4

**Summary:**

The paper introduces the DynaMer Adapter, an architecture that merges tokens from both general and medical pre-trained Vision Transformers (ViTs) to improve performance on medical imaging tasks. The DynaMer model leverages a Gated Mixture-of-Experts (MoE) Adapter for dynamically selecting relevant features and employs a layer-wise skipping router to optimize computational resources. Experimental results on the Med-VTAB benchmark indicate that DynaMer performs well, especially on few-shot and out-of-distribution tasks, suggesting its potential in specialized medical image applications.

**Strengths:**

Innovative Architecture: The gated MoE Adapter is a novel approach to merging features from domain-specific and general-purpose ViTs, potentially improving adaptation to complex medical tasks.

Effective on Benchmark Tasks: The model demonstrates state-of-the-art performance on Med-VTAB, particularly excelling in challenging medical scenarios with limited data.

Comprehensive Experiments: Extensive benchmarking and ablation studies were conducted, allowing for a detailed understanding of the architecture's components.

**Weaknesses:**

Efficiency Focus Unsubstantiated: Despite claims of computational efficiency, there is no direct comparison of inference or training time; only the parameter count is reported. Given that two full image backbones are used, inference time could increase substantially, undermining the claim of efficiency.

Marginal Performance Gain: The architecture, while sophisticated, yields limited improvements, making its complexity appear disproportionate to the performance gains observed.

Limited Baseline Comparison: Key baseline methods, such as directly fine-tuning general domain or medical-specific ViTs with Parameter-Efficient Fine-Tuning (PEFT) techniques, are not included. This omission raises concerns about the method’s effectiveness relative to simpler, more straightforward approaches.

**Questions:**

Ablation on Gated Mixture-of-Experts: Is the ablation study on the gating mechanism performed using the same model with gating modifications, or are separate models fine-tuned for each gating variation?

Comparison with Natural Baselines: Why were simpler baselines—such as direct fine-tuning of the general or medical domain ViT using PEFT—not included? If DynaMer does not outperform these baselines, its complex design may not be justified.

Explanation of Baseline Methods: Baselines such as VPT, GaPT, and LSPT are referenced, but there is no description of their differences. A simple explanation and comparison with DynaMer would enhance clarity and contextualize the model’s improvements.

---

### Meta-Review · Area_Chair_W5Wy · 2024-12-22

**Metareview:**

The paper introduces the DynaMer Adapter, a new architecture for integrating tokens from general and medical pre-trained Vision Transformers (ViTs) to enhance medical imaging tasks. While it demonstrates state-of-the-art performance on Med-VTAB, especially in few-shot and out-of-distribution settings, reviewers question its incremental contributions, as performance improvements are marginal (0.5%-1%) and insufficiently differentiated from existing MoE methods. The final average rating is below the acceptance threshold.

**Additional Comments On Reviewer Discussion:**

During the discussion period, reviewers highlighted that the paper lacks critical baseline comparisons, such as direct fine-tuning of general or medical ViTs or task-specific medical models, and fails to substantiate its computational efficiency claims with metrics like FLOPs or inference time. Additionally, key components, including the skipping router and gating mechanism, are insufficiently explained, and the advantages of combining general and medical models are not convincingly demonstrated. In their rebuttal, the authors provided additional experiments that partially addressed these concerns; however, the reviewers' enthusiasm remained limited. While two reviewers, satisfied to some extent, assigned final ratings of 5 and 6, this still reflects lingering doubts about the method's effectiveness and a lack of recognition of its broader significance.

---

### Decision · Program_Chairs · 2025-01-22

Reject